# Mapping the microRNA-mediated crosstalk between insulin resistance and Alzheimer's disease: A computational genomic insight

Tehniat Faraz Ahmed[1], Affan Ahmed[2], Zeba Haque[3], Muhammad Bilal Azmi●[4], Jawwad Us Salam[5], Farina Hanif●[3]*

1 Department of Biochemistry, Dow International Dental College, Dow University of Health Sciences, Karachi, Pakistan, 2 Dow Medical College, Dow University of Health Sciences, Karachi, Pakistan, 3 Department of Biochemistry, Dow International Medical College, Dow University of Health Sciences, Karachi, Pakistan, 4 Computational Biochemistry Research Laboratory, Department of Biochemistry, Dow Medical College, Dow University of Health Sciences, Karachi, Pakistan, 5 Department of Neurology, Dow University Hospital, Dow University of Health Sciences, Karachi, Pakistan

* Farina.Hanif@duhs.edu.pk

## Abstract

Insulin resistance (IR) and Alzheimer's disease (AD) share overlapping molecular mechanisms, but the precise link between these conditions remains unclear. MicroRNAs, as post-transcriptional regulators of gene expression, may mediate this connection by targeting genes involved in both pathways. In this study, we employed a multi-step bioinformatics approach to identify microRNAs that simultaneously regulate genes associated with IR and AD. Twenty key IR-related genes were selected from the literature, and their microRNA regulators were predicted using five computational tools. These predictions were validated using experimentally supported databases (TarBase and miRTarBase), and each miRNA–gene interaction was scored. Sixteen high-confidence microRNAs were shortlisted based on cumulative prediction and validation scores. These microRNAs were then analyzed for their interactions with AD pathway genes via KEGG pathway analysis. The AD-related target genes were further processed through protein–protein interaction network analysis using STRING and hub gene identification via Cytoscape. Functional enrichment of these hub genes using Gene Ontology and KEGG analysis revealed their involvement in shared biological processes, including apoptosis, insulin signaling, glucose metabolism, and neuroinflammation. Prominent candidates such as miR-7-5p, miR-106b, miR-424-5p, and miR-15a were identified. These results suggest that a subset of microRNAs may serve as critical molecular links between IR and AD, offering potential targets for early diagnosis and intervention.

**Data availability statement:** All relevant data are within the manuscript and its Supporting Information files.

**Funding:** The author(s) received no specific funding for this work.

**Competing interests:** The authors have declared that no competing interests exist.

**Abbreviation:** AD: Alzheimer's Disease; GO: Gene Ontology; IR: Insulin Resistance; KEGG: Kyoto Encyclopedia of Genes and Genomes; LOAD: Late-onset Alzheimer's Disease; MetS: Metabolic Syndrome;T2DM: Type 2 Diabetes Mellitus.

## Introduction

Alzheimer's disease (AD) is the most common form of dementia which accounts for 60–80% of all dementia cases globally. Its prevalence is increasing, particularly in lower-middle-income countries, which now contribute to over 60% of new cases [1]. Several known risk factors contribute to AD, with age being the most significant. Family history and genetics also play a role, particularly the well-characterized *APOE4* (ID:348) allele in late-onset AD (LOAD). Lifestyle factors like physical inactivity, poor diet, and smoking further increase the risk. Chronic neuroinflammation and traumatic brain injury are additional contributors. In recent years, the role of comorbid conditions, such as cardiovascular disease, Type 2 Diabetes Mellitus (T2DM), and Metabolic Syndrome (MetS), has been further emphasized as an established risk factors for AD [2,3]. A recent study highlighted that MetS-related comorbidities account for 74–81% of all LOAD cases [4]. Many of these modifiable and genetic risk factors for AD, such as T2DM, other components of MetS, and the APOE4 allele, are fundamentally linked to Insulin Resistance (IR).

IR occurs when cells in the body, such as muscle, fat, and liver cells, become less responsive to insulin, resulting in increased insulin production by the pancreas. Over time, this causes elevated blood glucose levels, which triggers inflammation, oxidative stress, and vascular damage, causing several conditions, such as cardiovascular disorders, obesity, T2DM, and MetS [5,6]. In addition to the above conditions, recent meta-analyses have provided substantial evidence of the link between cardiometabolic risks and cognitive decline [7,8].

In AD and T2DM, evidence suggests that IR induces metabolic dysfunction, which leads to bioenergetic failure [9]. IR and diabetes are also associated with elevated tau levels in AD [10]. The *APOE* gene, particularly its *E4* allele (*APOE4*), is the strongest known genetic risk factor for LOAD. APOE4 is associated with IR, a hypometabolic state, and reduced cerebral blood flow, all of which contribute to cognitive decline [9]. Although considerable evidence suggests a connection between IR and AD, the precise mechanism by which IR contributes to AD pathology remains unclear [11]. A recent genome-wide association study between AD and T2DM found no genetic overlap between the two conditions [12]. This finding highlights the need for alternative approaches to understanding the link between IR and AD, with epigenetic regulation emerging as a promising area of investigation. One such approach involves microRNAs (miRNAs) that regulate gene expression at the post-transcriptional level.

MiRNAs are small noncoding RNAs, typically 20–22 nucleotides long, that bind to messenger RNAs to block their translation [13,14]. By controlling gene expression, miRNAs can fine-tune various physiological and pathological processes, including cell differentiation proliferation, metabolism, apoptosis, and the molecular mechanisms of diseases [15]. Aberrant miRNA expression is also associated with several pathological conditions. Since miRNAs can simultaneously regulate multiple genes across different biological pathways, they are potential key players in the co-occurrence of conditions like IR, T2DM, and AD [16].

Research has shown the role that miRNAs may play in AD pathology [17,18]. Their function in regulating important proteins, like amyloid precursor protein

and β-secretase, along with their role in influencing phosphorylation of tau proteins, may contribute significantly to the characteristic AD changes of amyloid-β plaques and tau tangles respectively [19–22]. Dysregulation of miRNAs also affects neuroinflammation, microglial hyperactivation, and macrophage polarization in the brain, all of which are central to AD progression [23,24]. Furthermore, the role of miRNAs in the consequences of IR, such as T2DM and MetS, is critical, as they regulate key genes involved in glucose homeostasis, lipid metabolism, and inflammatory responses, all of which contribute to the progression and complications of these conditions [25–28]. Despite extensive research, the molecular mechanisms linking IR and AD remain poorly defined, representing a key gap in our understanding of AD pathophysiology. There is a need for hypothesis driven, systematic investigation to elucidate the role of miRNAs in connecting metabolic dysfunction to neurodegenerative changes. Such an understanding could provide valuable insights for identifying diagnostic biomarkers and developing new therapeutic strategies for AD.

This study aims to identify miRNAs that are common to both IR and AD pathways and to characterize their potential biological roles in linking metabolic and neurodegenerative processes. We hypothesize that certain miRNAs serve as common regulatory nodes that concurrently influence both IR and AD pathways, thereby mediating the crosstalk between metabolic dysfunction and neurodegeneration.

We began by evaluating miRNAs that target genes associated with the IR pathway using an integrative approach that combined several prediction algorithms with validated experimental datasets. The top-ranking miRNAs from this initial screen were analyzed for their potential involvement in AD-related biological pathways. Through this analysis, we identified key proteins playing central roles in the AD network and reassessed miRNAs based on their predicted interactions with these critical targets. The final set of miRNAs represents candidates that may influence both IR and neurodegenerative processes in the brain. These findings highlight the need for further exploration of the expression patterns of these miRNAs in individuals affected by AD and IR-linked conditions.

## Methodology

### Selection of insulin resistance pathway genes for analysis

A focused, literature-based approach was used to identify genes implicated in IR. To keep the analysis manageable and computationally feasible, we prioritized a concise set of functionally distinct, consistently reported, and biologically well-established modulators of the IR pathway at both proximal and distal levels [5,29–35].

Genes were included if they met the following criteria:

(i)  They were recurrently reported across multiple peer-reviewed studies or reviews as key regulators of IR, and

(ii)  They represented functionally distinct components of the IR network, encompassing receptor activation, intracellular signal transduction, and metabolic effector regulation.

Because this study specifically targeted the canonical insulin signaling cascade and its downstream metabolic effectors, IGF-related genes were excluded to maintain pathway specificity. Similarly, only the metabolic arm of insulin signaling (the PI3K–AKT–mTOR axis) was analyzed, whereas the mitogenic arm (RAS–MAPK) was excluded.

Several well-known modulators that influence IR—such as phosphatases, inflammatory and feedback regulators, and transcriptional or metabolic sensors were also excluded. These genes affect IR indirectly or through tissue-specific mechanisms rather than through direct propagation of insulin receptor signaling.

We acknowledge that this targeted strategy necessarily omits some genes implicated in IR. However, the objective of this study was not to generate an exhaustive catalogue of all IR-associated genes. While this introduces a degree of selection bias, the curated panel captures the principal molecular nodes of IR and provides a biologically meaningful framework for exploring miRNA-mediated links between metabolic and neurodegenerative pathways.

The final panel comprised 20 genes spanning the major levels of IR: receptor activation, adaptor and kinase intermediates, and downstream metabolic effectors. Full gene details, functions and rationale for inclusion or exclusion are provided in Supplementary S1 File.

## Identification of microRNAs targeting genes involved in IR

To identify miRNAs that target the identified genes, we input the compiled list of genes into 5 algorithms: TargetScan Human v8.0 (https://www.targetscan.org/), miRDB v6.0 (https://mirdb.org/), miRmap (https://mirmap.ezlab.org/app), DIANA-microT Webserver 2023 (https://dianalab.e-ce.uth.gr/microt_webserver) and miRTar2GO (http://www.mirtar2go.org/index.html). Only predicted miRNA targets in Homo sapiens were considered. Each gene was individually analyzed, prioritizing conserved targets. Poorly conserved targets were excluded from the analysis. In cases where multiple transcripts existed for a gene, we focused on the most prevalent transcript for the analysis. A predicted target with a target score > 80 is most likely real (https://mirdb.org/faq.html); therefore, a target score cut-off of 0.80 was kept in miRDB, miRmap and DIANA-microT. Fig 1 displays a flowchart of the methodology used.

We validated the computationally predicted interactions using experimental data from two databases: TarBase v9.0 (https://dianalab.e-ce.uth.gr/tarbasev9) and miRTarBase 2022 (https://mirtarbase.cuhk.edu.cn/). Although experimental verification is essential, such databases are few and do not include all the interactions.

## Scoring of microRNAs based on their predicted interaction with IR pathway genes

To quantitatively assess the relevance of each miRNA in regulating IR pathway genes, a scoring system was implemented based on both computational predictions and experimental validation. For each miRNA–gene pair, a predicted score (P-score) was determined by dividing the number of algorithms (denoted as n) that identified the interaction by the total number of tools used in this study (n/5). Thus, the P-score ranged from 0.2 (predicted by one tool) to 1.0 (predicted by all five prediction tools). An experimental score (E-score) was assigned to account for the experimental evidence. Interactions validated in either TarBase v9.0 or miRTarBase 2022 received one additional point; otherwise, the E-score was zero. The cumulative score (C-score) for each miRNA–gene interaction was calculated as the sum of the P-score and E-score, resulting in a possible range of 0.2–2.0. To rank miRNAs according to their overall predicted impact on the IR pathway, a total score (T-score) was calculated for each miRNA. This score was derived by summing the C-scores of a given miRNA across all 20 selected IR-related genes and dividing the total by 20 using the following formula (equation 1):

$$T-score = \frac{1}{20} \sum_{i=1}^{20} \left( \frac{n_i}{5} + E-score_i \right)$$

where *i* represents the index of each gene.

Based on this scoring framework, miRNAs were ranked according to their overall regulatory potential. The T-score integrates both the breadth of predicted regulation (number of IR genes targeted) and the strength of supporting evidence (cross-tool and experimental validation). Consequently, higher T-scores identify miRNAs with broader and more consistent involvement across multiple nodes of the insulin-signaling network, suggesting potential network-level regulatory roles rather than isolated single-gene effects. A miRNA was retained in the final list if its average T-score across all 20 insulin-resistance–related genes exceeded 1.0. We selected this threshold because a miRNA with this score either showed, on average, experimental evidence for targeting the genes or was consistently predicted by all computational tools. Thus, it selected candidates with both strong prediction confidence and broad regulatory relevance, while excluding weak or isolated predictions. Notably, miRNAs predicted by a single algorithm but validated experimentally across all genes achieved a minimum T-score of 1.2.

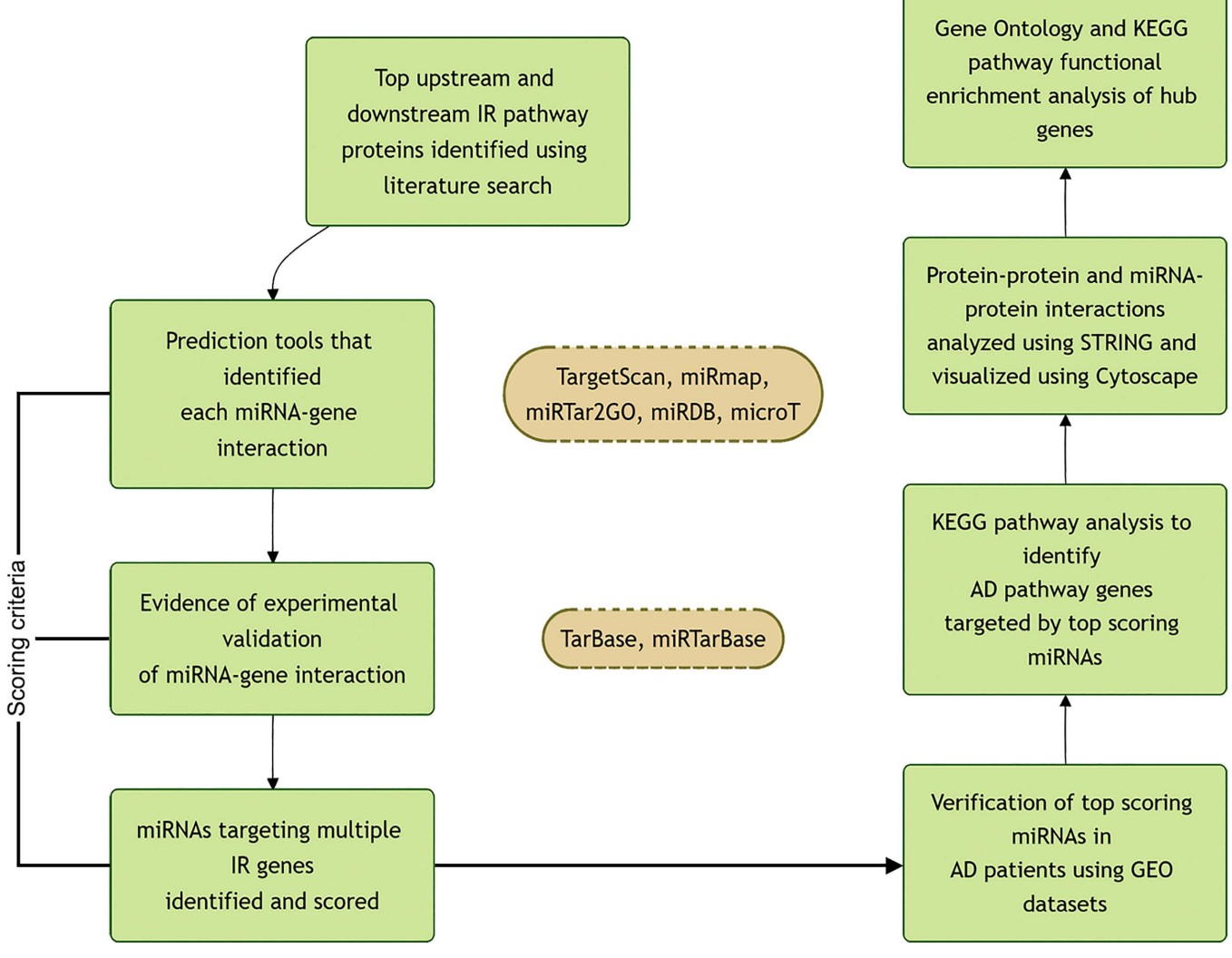

**Fig 1. Workflow of the study.**

## Validation of shortlisted microRNAs in AD patients

Global miRNA expression profiles were obtained from two publicly accessible Gene Expression Omnibus (GEO) datasets: GSE46579 and GSE48552. The former dataset contains blood miRNA expression data from 48 AD patients and 22 controls, while the latter dataset contains miRNA expression data of 6 AD patients and 6 controls from their prefrontal cortex. Both datasets used Illumina HiSeq 2000 platforms.

For both datasets, pre-processed count matrices provided by GEO were downloaded and imported into R for differential expression analysis. Filtering of low-abundance miRNAs, normalization, and dispersion estimation were performed using the edgeR package. Differential expressions were assessed using a negative binomial generalized linear model with false discovery rate correction. Significantly dysregulated miRNAs were defined using a threshold of <0.05. Differentially expressed miRNAs identified from both datasets were compared with the high-confidence candidate miRNAs obtained from our IR-associated computational screen to identify overlapping signatures relevant to the IR–AD axis.

### Identification of hub proteins in the AD pathway targeted by top-scoring microRNAs

To explore the functional relevance of the top-scoring miRNAs, we identified their potential targets within the AD pathway. KEGG pathway enrichment analysis was performed using miRPath v4.0 (http://microrna.gr/miRPathv4/), incorporating both computational predictions and experimentally validated miRNA-gene interactions. This allowed us to generate a list of AD-associated genes regulated by the shortlisted miRNAs. Following the analysis, all identified genes were consolidated into a non-redundant list of genes. To investigate their functional connectivity, protein–protein interaction analysis was conducted using the STRING database, applying a high-confidence interaction threshold (score ≥ 0.7). The resulting protein–protein interaction network was imported into Cytoscape v3.10.2, and the CytoHubba plug-in was used to determine the most central proteins within the network based on the degree algorithm.

The top ten proteins with the highest degree scores were classified as hub proteins in the AD pathway. These hub proteins, owing to their key roles in network connectivity, were selected for subsequent analyses to further refine miRNA prioritization. We also generated miRNA–gene interaction diagrams for the identified AD hub genes and for the IR genes using Cytoscape.

### Functional enrichment analysis of key genes

To investigate the biological significance of genes positioned at the intersection of AD and IR pathways, we conducted a functional enrichment analysis focusing on ten AD-associated hub genes previously identified as targets of high-ranking IR-related miRNAs. These genes are proposed to function as crucial molecular mediators linking IR-related miRNA activity to the pathophysiology of AD. Enrichment analysis was performed using ShinyGO v0.82 (https://bioinformatics.sdstate.edu/go/), a web-based tool for gene set annotation. Ensembl gene identifiers for the selected hub genes were submitted to evaluate overrepresented pathways in both the Kyoto Encyclopedia of Genes and Genomes (KEGG) and Gene Ontology (GO) domains, including the Biological Process, Molecular Function, and Cellular Component categories.

To ensure statistical rigor, a false discovery rate threshold of <0.05 was applied, and only pathways containing between 2 and 5000 genes were considered for analysis. The top 10 significantly enriched pathways in each category were identified and ranked based on the number of hub genes involved, providing insights into the functional roles these genes may play in the convergence of AD and IR mechanisms.

## Results

### Scoring of key microRNAs in the insulin resistance pathway

Based on our selection criteria, we identified 7 upstream, and 13 downstream regulators of IR pathway. The genes encoding these proteins, along with their respective symbols, are listed in Table 1. By entering the names of these genes individually into five computational algorithms, we compiled a list of miRNAs predicted to target each gene. For each gene, we then combined the miRNA lists from all tools and removed any duplicates. Table 2 lists the number of predicted interactions for each gene with miRNA according to the algorithm used.

Among the genes, *TSC1* had the highest number of predicted miRNA-gene interactions, with a total of 554, followed closely by *AKT3* with 529 interactions. In contrast, only 31 miRNAs were predicted to target the *TSC2* gene across all algorithms.

To validate these predictions, we cross-referenced our findings with two databases containing experimentally verified miRNA-gene interactions. The *GSK3B* gene had the highest number of experimentally verified miRNAs targeting it, with 309 interactions, while the *SLC2A4* gene had the fewest, with 60 verified interactions. After merging the computational predictions with the experimentally validated interactions and removing duplicates, *AKT3* emerged as the gene targeted by the most miRNAs (651), whereas *AKT1S1* had the fewest (170).

Following the compilation of the miRNA list, we scored each miRNA using the formulas outlined in the methodology section, calculating the P-, E-, C-, and T-scores. These scores are detailed in Supplementary S2 File.

Sixteen miRNAs with T-score greater than 1.0 were prioritized for downstream analyses (Table 3). Among these, miR-15a-5p had the highest score of 1.23. Additionally, miR-424-5p, miR-15b-5p, miR-16-5p, let-7d-5p, and miR-17-5p had

**Table 1. Selected genes from insulin resistance pathway.**

| Genes | Gene Symbol | Entrez ID | Protein coded |
|---|---|---|---|
| *Upstream regulators* | | | |
| Insulin Receptor | INSR | 3643 | Insulin receptor protein |
| Insulin Receptor Substrate 1 | IRS1 | 3667 | Insulin Receptor Substrate 1 |
| Insulin Receptor Substrate 2 | IRS2 | 8660 | Insulin Receptor Substrate 2 |
| Phosphoinositide-3-kinase regulatory subunit 1 | PIK3R1 | 5295 | p85α regulatory subunit of phosphoinositide 3-kinase |
| Phosphoinositide-3-kinase regulatory subunit 2 | PIK3R2 | 5296 | p85β regulatory subunit of PI3K |
| AKT serine/threonine kinase 2 | AKT2 | 208 | protein kinase B beta |
| AKT serine/threonine kinase 3 | AKT3 | 10000 | protein kinase B gamma |
| *Downstream regulators* | | | |
| Glycogen synthase kinase 3 beta | GSK3B | 2932 | Glycogen Synthase Kinase-3 beta |
| Pyruvate dehydrogenase kinase 1 | PDK1 | 5163 | Pyruvate dehydrogenase kinase isozyme 1 |
| Forkhead box O1 | FOXO1 | 2308 | Forkhead box protein O1 |
| Mechanistic target of rapamycin kinase | MTOR | 2475 | Serine/threonine-protein kinase mTOR |
| Tuberous sclerosis complex subunit 1 | TSC1 | 7248 | Hamartin protein |
| Tuberous sclerosis complex subunit 2 | TSC2 | 7249 | Tuberin protein |
| AKT1 Substrate 1 | AKT1S1 | 84335 | Proline-rich AKT1 Substrate |
| Sterol Regulatory Element Binding Transcription Factor 1 | SREBF1 | 6720 | Sterol Regulatory Element Binding Protein 1c |
| Phosphodiesterase 3B | PDE3B | 5140 | Phosphodiesterase 3B protein |
| Acetyl-CoA Carboxylase Alpha | ACACA | 31 | acetyl-CoA carboxylase alpha (ACC-alpha) enzyme |
| Acetyl-CoA Carboxylase Beta | ACACB | 32 | Acetyl-CoA Carboxylase Beta (ACACB) enzyme |
| Alpha/Beta Hydrolase Domain-Containing Protein 15 | ABHD15 | 116236 | Abhydrolase domain containing 15 |
| Solute Carrier Family 2 Member 4 | SLC2A4 | 6517 | Facilitated Glucose Transporter, Member 4 |

**Table 2. Frequency of microRNA-gene interactions by each gene.**

| Gene | TargetScan | miRDB | miRmap | microT | miRTar2GO | Predicted total | TarBase | miRTarBase | Experimental total | Overall total |
|---|---|---|---|---|---|---|---|---|---|---|
| INSR | 39 | 55 | 325 | 124 | 147 | 528 | 187 | 4 | 188 | 628 |
| IRS1 | 50 | 76 | 100 | 210 | 4 | 315 | 247 | 42 | 276 | 524 |
| IRS2 | 39 | 49 | 213 | 107 | 10 | 316 | 227 | 28 | 244 | 492 |
| PIK3R1 | 53 | 124 | 277 | 199 | 10 | 436 | 230 | 55 | 266 | 609 |
| PIK3R2 | 8 | 15 | 102 | 27 | 8 | 136 | 99 | 8 | 102 | 229 |
| AKT2 | 22 | 8 | 224 | 33 | 33 | 287 | 104 | 31 | 123 | 369 |
| AKT3 | 74 | 89 | 291 | 246 | 7 | 529 | 154 | 53 | 190 | 651 |
| GSK3B | 56 | 100 | 11 | 332 | 6 | 394 | 275 | 64 | 309 | 620 |
| PDK1 | 9 | 27 | 28 | 375 | 1 | 405 | 85 | 19 | 97 | 482 |
| FOXO1 | 38 | 94 | 298 | 165 | 9 | 413 | 138 | 69 | 183 | 540 |
| MTOR | 11 | 34 | 83 | 52 | 4 | 128 | 208 | 23 | 215 | 324 |
| TSC1 | 65 | 79 | 342 | 140 | 152 | 554 | 180 | 20 | 191 | 632 |
| TSC2 | 0 | 0 | 9 | 13 | 9 | 31 | 160 | 3 | 161 | 187 |
| AKT1S1 | 0 | 0 | 65 | 19 | 0 | 79 | 96 | 4 | 96 | 170 |
| SREBF1 | 7 | 6 | 40 | 19 | 24 | 91 | 173 | 42 | 202 | 276 |
| PDE3B | 81 | 73 | 132 | 107 | 132 | 373 | 109 | 6 | 110 | 414 |
| ACACA | 0 | 13 | 191 | 63 | 0 | 232 | 267 | 40 | 295 | 498 |
| ACACB | 0 | 24 | 151 | 55 | 19 | 209 | 59 | 15 | 73 | 270 |
| ABHD15 | 16 | 9 | 194 | 33 | 0 | 224 | 33 | 106 | 137 | 328 |
| SLC2A4 | 22 | 24 | 134 | 37 | 18 | 193 | 33 | 31 | 60 | 227 |

**Table 3. C and T scores of the microRNA-genes interactions.**

| miRNA | INSR | IRS1 | IRS2 | PIK3R1 | PIK3R2 | AKT2 | AKT3 | GSK3B | PDK1 | FOXO1 | MTOR | TSC1 | TSC2 | AKT1S1 | SREBF1 | PDE3B | ACACA | ACACB | ABHD15 | SLC2A4 | Total | T-Score |
|---|---|---|---|---|---|---|---|---|---|---|---|---|---|---|---|---|---|---|---|---|---|---|
| miR-15a-5p | 2 | 1.4 | 1.2 | 1.8 | 1 | 1.2 | 1.8 | 1.2 | 1 | 1.4 | 1 | 1.6 | 1 | 1 | 1 | 1.6 | 1 | 1.4 | 0 | 1 | 24.6 | 1.23 |
| miR-424-5p | 2 | 1.6 | 1.4 | 1.8 | 1 | 1.2 | 1.8 | 1.4 | 1 | 1.4 | 1 | 1.6 | 1 | 1 | 1 | 1.6 | 1 | 1.4 | 0 | 0 | 24.2 | 1.21 |
| miR-15b-5p | 2 | 1.4 | 1.2 | 1.8 | 1 | 1.2 | 1.8 | 1.2 | 0 | 1.4 | 1 | 1.6 | 1 | 1 | 1 | 1.6 | 1 | 1.4 | 1 | 0 | 23.6 | 1.18 |
| miR-16-5p | 2 | 1.4 | 1.2 | 1.8 | 1 | 0.2 | 1.8 | 1.2 | 0 | 1.4 | 1 | 1.6 | 1 | 1 | 1 | 1.6 | 1 | 1.4 | 1 | 0 | 22.6 | 1.13 |
| let-7d-5p | 1.6 | 1 | 1.6 | 1 | 1 | 1.2 | 1 | 1 | 1 | 1 | 1 | 1.6 | 1 | 1 | 1 | 1.2 | 1 | 1 | 1 | 1 | 22.2 | 1.11 |
| miR-17-5p | 1.6 | 1 | 1 | 1.4 | 1 | 1 | 1.4 | 1 | 1 | 1 | 1 | 1.2 | 1 | 1 | 1 | 2 | 1 | 0 | 1.2 | 1.4 | 22.2 | 1.11 |
| let-7i-5p | 1.8 | 1 | 1.6 | 1 | 1 | 1.4 | 1 | 1 | 1 | 1 | 1 | 1.6 | 1 | 1 | 1 | 1.2 | 1 | 1 | 1 | 0 | 21.6 | 1.08 |
| let-7-5p | 1.4 | 1.8 | 1.8 | 1 | 1 | 1 | 1.2 | 1.2 | 0.2 | 1 | 1.4 | 1.2 | 1 | 1 | 1 | 0.2 | 1 | 1 | 1 | 1.2 | 21.6 | 1.08 |
| miR-27a-3p | 1.8 | 1.4 | 1 | 1 | 0 | 1 | 1 | 1.4 | 1.4 | 1.6 | 1 | 2 | 1 | 1 | 1 | 1.6 | 1.2 | 0 | 1 | 0 | 21.4 | 1.07 |
| miR-7a-5p | 1.8 | 1 | 1.6 | 1 | 1 | 1.2 | 1 | 1 | 1 | 1 | 1 | 1.6 | 1 | 1 | 1 | 1.2 | 1 | 1 | 1 | 0 | 21.4 | 1.07 |
| miR-7e-5p | 1.8 | 1 | 1.6 | 1 | 1 | 1.2 | 1 | 1 | 1 | 1 | 1 | 1.6 | 1 | 1 | 1 | 1.2 | 1 | 1 | 1 | 0 | 21.4 | 1.07 |
| let-7b-5p | 1.8 | 1 | 1.6 | 1 | 1 | 1.2 | 1 | 1 | 1 | 1 | 1 | 1.4 | 1 | 1 | 1 | 1.2 | 1 | 1 | 1 | 0 | 21.2 | 1.06 |
| miR-106b-5p | 1.4 | 1 | 1 | 1.4 | 1 | 1 | 1.4 | 1 | 1 | 1 | 1 | 1 | 1 | 0 | 1 | 1.8 | 1 | 0 | 1.2 | 1.4 | 20.6 | 1.03 |
| let-7c-5p | 1.8 | 1 | 1.6 | 1 | 1 | 1.2 | 1 | 1 | 1 | 1 | 1 | 1.6 | 1 | 1 | 1 | 0.2 | 1 | 1 | 1 | 0 | 20.4 | 1.02 |
| miR-20b-5p | 1.6 | 1 | 1 | 1.4 | 1 | 0 | 1.4 | 1 | 1 | 1 | 1 | 1.2 | 1 | 1 | 1 | 2 | 0 | 0 | 1.2 | 1.4 | 20.2 | 1.01 |
| miR-93-5p | 1.6 | 1 | 1 | 1.4 | 1 | 0 | 1.4 | 1 | 1.2 | 1 | 1 | 1.2 | 1 | 0 | 1 | 1.6 | 1 | 0 | 1.4 | 1.4 | 20.2 | 1.01 |

scores exceeding 1.10. The overlap of predicted miRNAs across the different computational algorithms is shown in Fig 2, and the miRNA–IR gene interaction diagram is presented in Fig 3. The thickness and color of the edges represent the strength of each interaction.

### Validation of candidate microRNAs in AD patients

We performed differential expression analysis on miRNome data from two publicly available GEO datasets and examined whether our 16 high-confidence candidate miRNAs appeared among the dysregulated miRNAs. Fourteen of the short-listed miRNAs were dysregulated in at least one dataset. Eleven of these miRNAs showed significant differential expression in GSE48552. Hsa-let-7e-5p was the only miRNA consistently dysregulated across both datasets. Two miRNAs (hsa-let-7b-5p and hsa-miR-20b-5p) were not dysregulated in either dataset. Supplementary S3 File lists all differentially expressed miRNAs identified in each dataset.

### Identification of AD-IR hub genes

The top 16 miRNAs identified earlier were further analyzed for their interactions with AD pathway genes. Using the KEGG pathway, we compiled a list of AD-related genes targeted by these miRNAs.

miR-16-5p targeted the highest number (146) of AD pathway genes, while miR-20b-5p targeted only 43 AD pathway-related genes (Table 4). Altogether, the miRNAs targeted 299 AD pathway-related genes. Supplementary S4 File contains the names of all these genes.

We then constructed a protein-protein interaction network using the compiled list of AD-related genes. The top 10 hub genes within this network, identified through degree centrality analysis are presented in Fig 4a. Among these, *GAPDH* (ID:2597) exhibited the highest number of interactions with other AD pathway genes, followed by *GSK3B* (ID:2932). The interaction network linking these hub genes with the shortlisted miRNAs from the IR pathway is shown in Fig 4b, highlighting potential post-transcriptional regulatory relationships between key AD and IR nodes. A simplified representation of the mechanistic roles of these top 10 hub genes within the AD KEGG pathway is provided in Fig 5.

### GO and KEGG analysis

Gene Ontology enrichment analysis revealed that the top molecular functions associated with the AD–IR hub genes included cytoskeletal protein binding, enzyme binding, and related activities (Fig 6). In the cellular component category, the most enriched terms were mitochondrion, cytoskeleton, and nucleoplasm. The predominant biological processes included cell death, regulation of catalytic activity, and the regulation of protein metabolic processes.

To further elucidate the functional roles of these genes, KEGG pathway analysis was conducted. As expected, a considerable proportion of the hub genes were enriched in the AD pathway. Additionally, several pathways related to infectious diseases, including hepatitis C and cytomegalovirus infection, were identified. Notably, neurodegenerative disease pathways were also featured prominently among the top results (Fig 7).

### Discussion

IR is a central feature of T2DM and a critical driver of metabolic dysfunction. It has emerged as a key contributor to the development and progression of AD. Impaired insulin signaling in the brain disrupts glucose metabolism, synaptic function, and neuronal survival, contributing to hallmark AD pathologies, such as tau hyperphosphorylation, amyloid β accumulation, and neuroinflammation [36]. Emerging evidence highlights the role of miRNAs as potential molecular intermediaries in this IR–AD connection, given their ability to fine-tune gene expression in both insulin signaling and neurodegenerative pathways. While previous studies have demonstrated the involvement of individual miRNAs in IR or AD, they have provided only fragmented insights into the broader regulatory network. The strength of our study lies in its integrative computational approach, which combines multiple databases, target prediction algorithms, and pathway enrichment analyses to

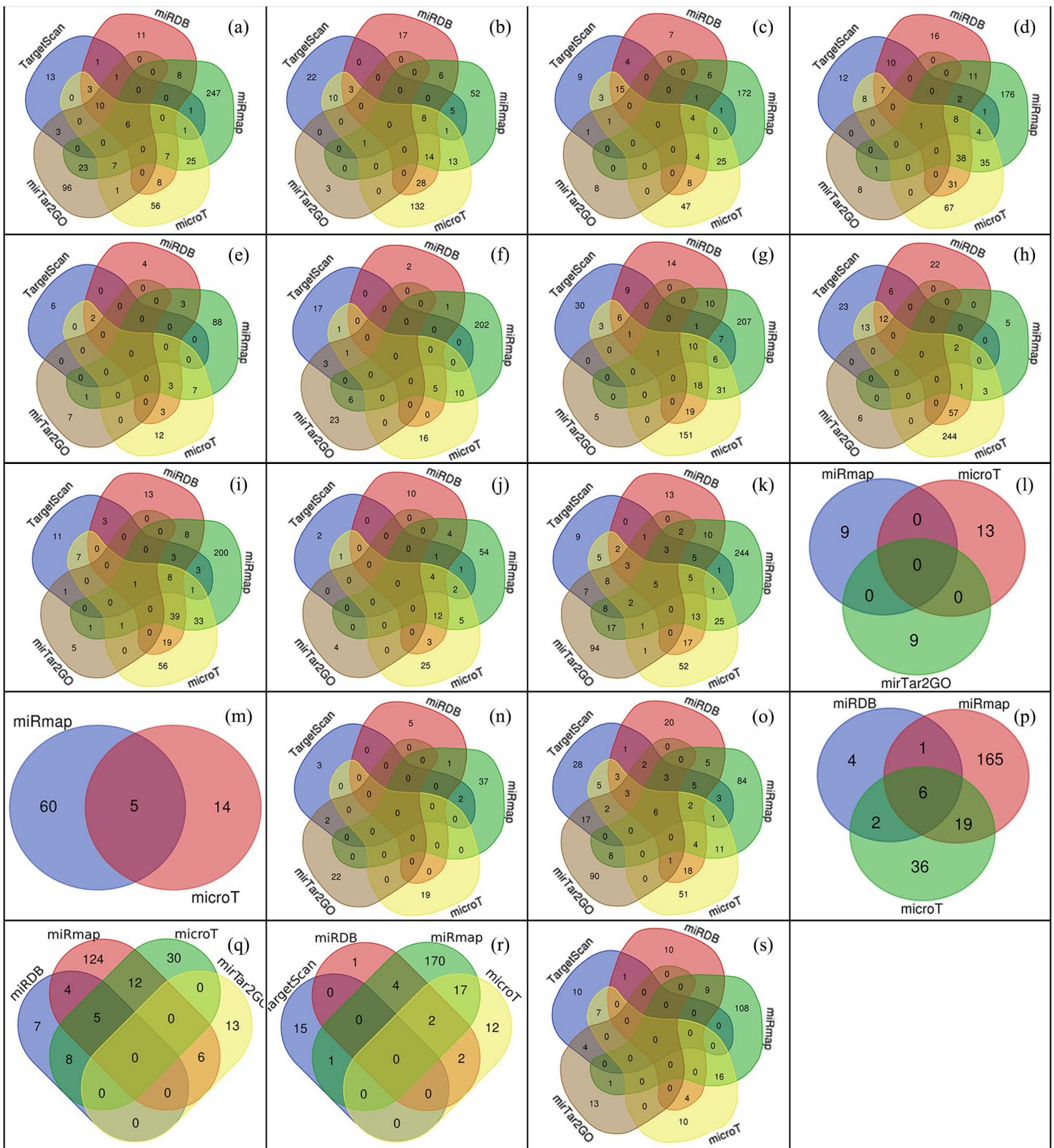

**Fig 2. Venn diagrams indicating number of overlapping miRNAs found by different experimental tools.** a) INSR b) IRS1 c) IRS2 d) PIK3R1 e) PIK3R2 f) AKT2 g) AKT3 h) GSK3B i) FOXO1 j) MTOR k) TSC1 l) TSC2 m) AKT1S1 n) SREBF1 o) PDE3B p) ACACA q) ACACB r) ABHD15 s) SLC2A4.

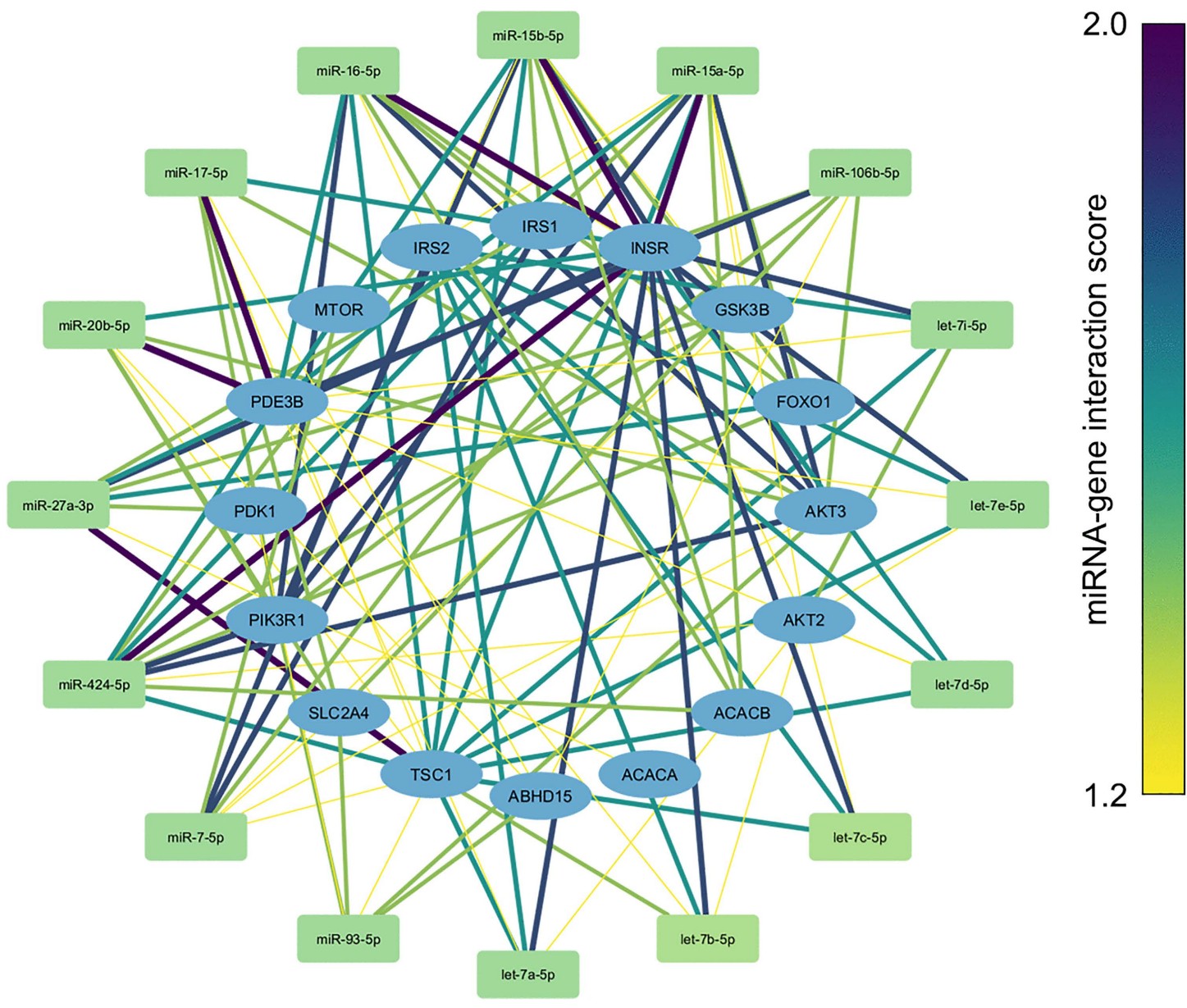

**Fig 3. miRNA-IR gene interaction diagram.**

construct a comprehensive miRNA–gene interaction map. This system-level view offers a deeper understanding of how miRNAs orchestrate overlapping molecular mechanisms in IR and AD, potentially revealing novel targets for early diagnosis or therapeutic intervention.

In silico miRNA prediction tools offer a cost-effective and powerful approach for dissecting complex regulatory networks in diseases with overlapping molecular pathways, such as IR and AD. These tools leverage binding site predictions, cross-species conservation, and experimentally supported gene–miRNA interactions to identify biologically relevant candidates for further study. To enhance confidence in our results, we employed a composite approach using five independent algorithmic platforms along with two curated experimental validation databases. This multi-step filtering strategy enabled us to prioritize miRNAs with consistent and high-confidence targets across both IR- and AD-related genes.

**Table 4. Frequency of AD pathway gene targeted by candidate microRNAs identified via KEGG pathway analysis.**

|  | TarBase | miRTarBase | TargetScan | Total |
|---|---|---|---|---|
| hsa-miR-16-5p | 122 | 65 | 45 | 146 |
| hsa-let-7b-5p | 90 | 46 | 28 | 120 |
| hsa-let-7a-5p | 83 | 27 | 28 | 113 |
| hsa-miR-27a-3p | 80 | 11 | 41 | 106 |
| hsa-miR-17-5p | 66 | 38 | 27 | 95 |
| hsa-miR-424-5p | 64 | 16 | 45 | 93 |
| hsa-miR-15a-5p | 63 | 31 | 45 | 92 |
| hsa-miR-7-5p | 77 | 27 | 27 | 91 |
| hsa-let-7c-5p | 58 | 24 | 28 | 88 |
| hsa-let-7e-5p | 56 | 34 | 28 | 88 |
| hsa-miR-15b-5p | 56 | 33 | 45 | 87 |
| hsa-miR-93-5p | 53 | 22 | 27 | 78 |
| hsa-let-7d-5p | 56 | 6 | 28 | 75 |
| hsa-let-7i-5p | 55 | 6 | 28 | 74 |
| hsa-miR-106b-5p | 46 | 25 | 27 | 72 |
| hsa-miR-20b-5p | 23 | 17 | 27 | 43 |
| Cumulative | 255 | 147 | 116 | 299 |

Our analysis identified several miRNAs with regulatory roles in insulin signaling, particularly through their targeting of core components, such as the insulin receptor and downstream signaling pathways. Using bioinformatics pipeline, we pinpointed a subset of miRNAs as strong candidates linking IR to AD-related genes. Notably, miR-93-5p, miR-17-5p, miR-16-5p, miR-20a-5p, and miR-106b-5p emerged as high-confidence regulators. These miRNAs have also been reported in previous studies as key modulators of transcriptional alterations associated with AD, reinforcing their potential role at the intersection of metabolic dysfunction and neurodegeneration [37].

MiR-106b-5p has been associated with an increased risk of AD based on changes in expression in whole blood [38,39]. This miRNA has also been reported to regulate GLUT4 expression, which is a central component of insulin-mediated glucose uptake [40]. Similarly, miR-93-5p, which also modulates GLUT4 expression, has been implicated in IR and was found to be significantly upregulated in extracellular vesicles derived from AD patients compared to healthy controls and patients with other forms of dementia [40–42]. Although previously investigated independently in IR and AD contexts, our study provides a novel perspective by proposing these miRNAs as potential mediators of AD in patients with underlying metabolic dysfunction.

Among the strongest candidates identified in our analysis, miR-424-5p directly binds to the 3′ untranslated region of INSR mRNA in hepatocytes, reducing INSR expression and impairing insulin signaling. Its upregulation has been linked to lipid-induced stress via the TGFβ-SMAD3 signaling pathway [43], and its dysregulation has been observed in insulin-resistant conditions such as polycystic ovary syndrome and obesity [44,45]. Notably, miR-424-5p, along with miR-93-5p and miR-15a-5p, has been reported to be significantly elevated in EVs from patients with AD [42]. Our identification of these miRNAs as regulators of insulin signaling in the context of AD pathology further supports their relevance at the intersection of metabolic and neurodegenerative processes.

Furthermore, the miR-15 family, including miR-15a and miR-15b, has been shown to suppress INSR expression in diet-induced obesity models and human cell lines. Elevated levels of these miRNAs have also been detected in the offspring of women with gestational diabetes mellitus, implicating them in the intergenerational transmission of IR and the increased risk of T2DM [46,47]. A study involving monozygotic twins discordant for T2DM also reported the differential expression of miR-15 family members [48]. Interestingly, miR-15a-5p and let-7i-5p, both of which we found relevant in our

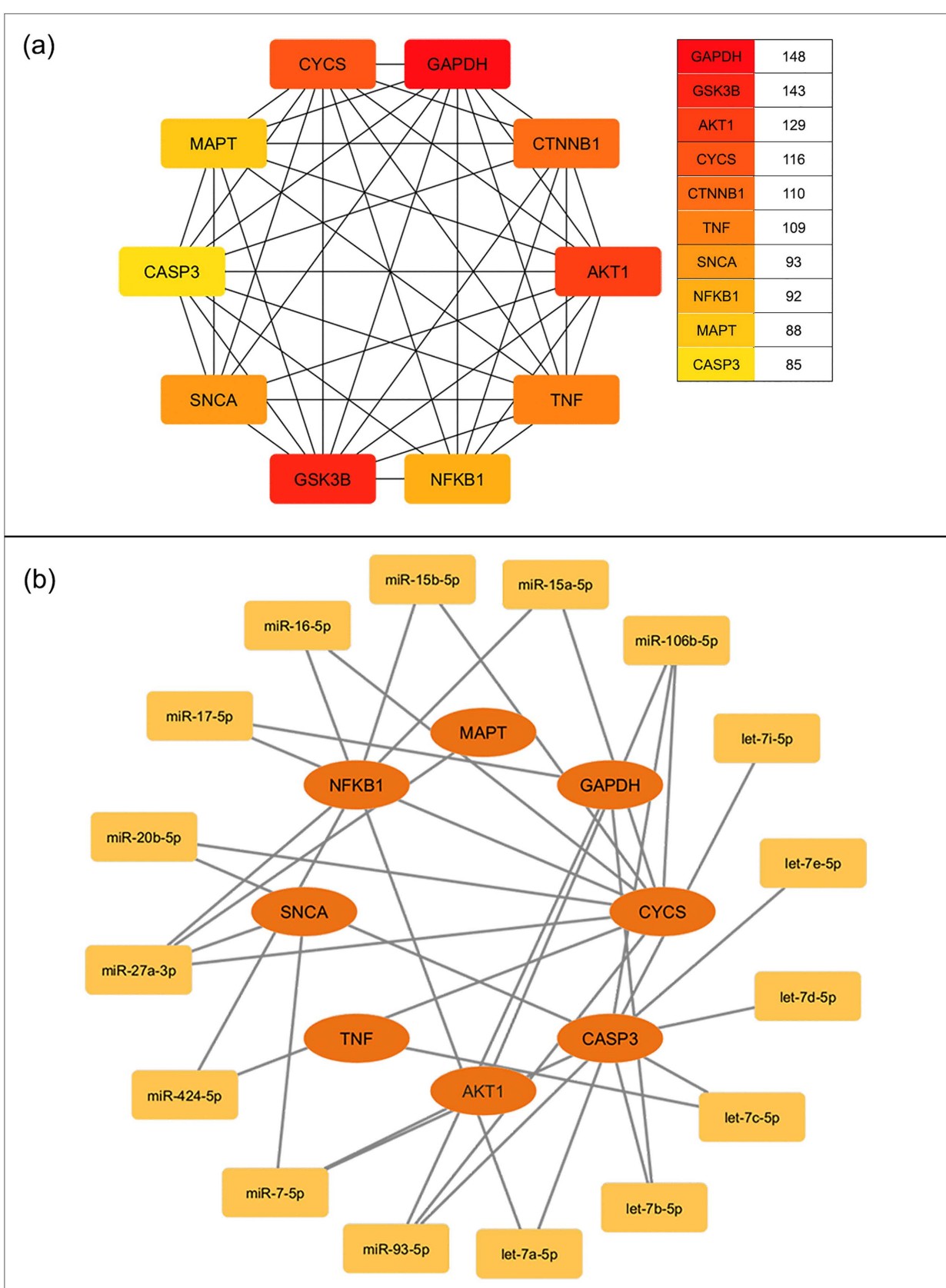

**Fig 4. KEGG-Derived Hub Genes and Their Regulatory microRNA Network.** (a) Top ten hub genes identified through AD KEGG pathway analysis using the degree method. (b) Interaction network linking these hub genes with the shortlisted microRNAs.

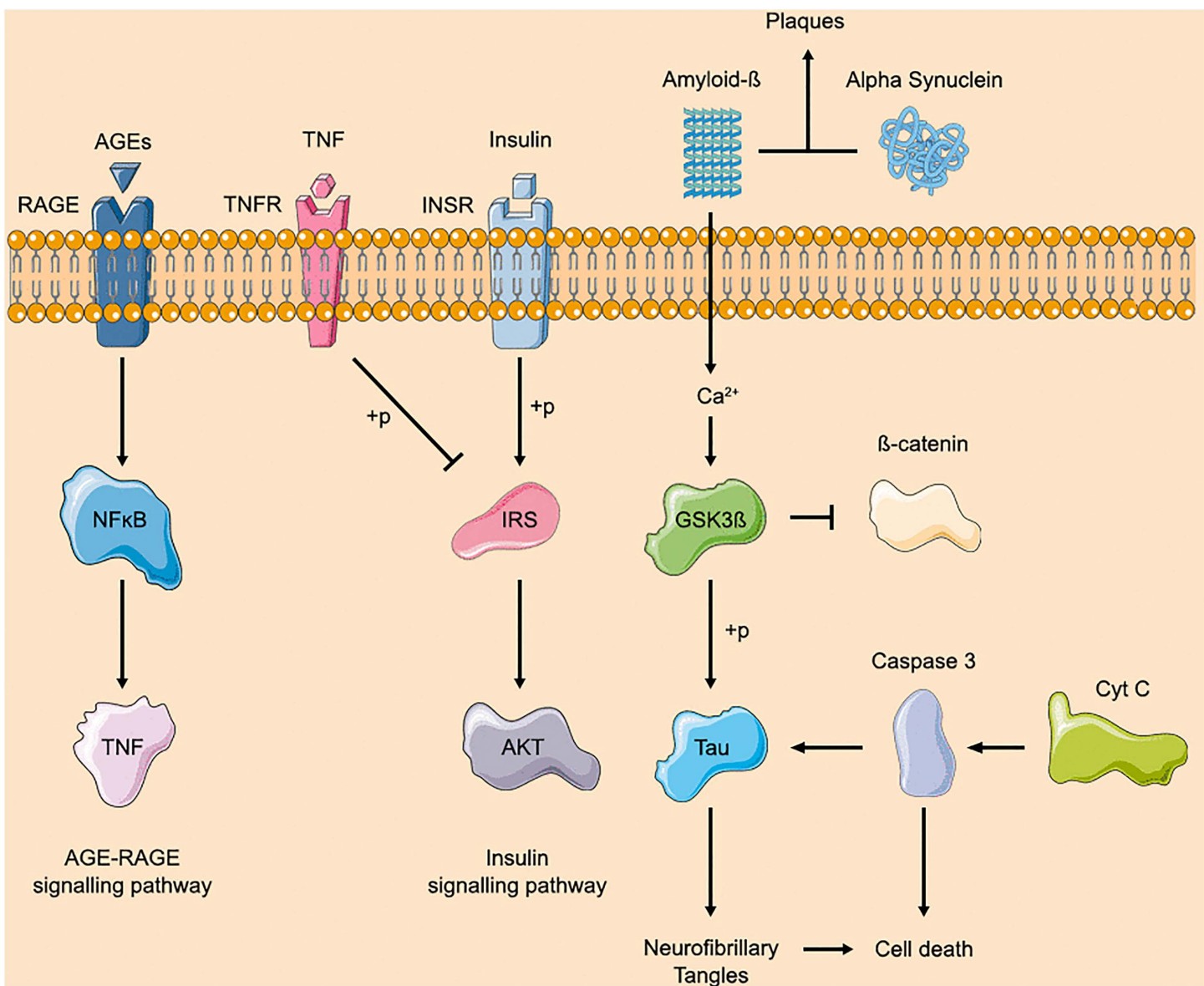

**Fig 5. Simplified KEGG pathway of top 10 AD hub genes identified through Cytoscape.**

pathway analysis, have previously been reported as differentially expressed in the blood and CSF of AD patients and are known to target important AD-related proteins such as amyloid precursor protein and BACE1 [49].

In our network analysis, miR-16-5p was predicted to target 146 AD-associated genes, highlighting its broad regulatory impact. This miRNA has also been shown to modulate insulin signaling proteins in myocytes and is associated with IR in

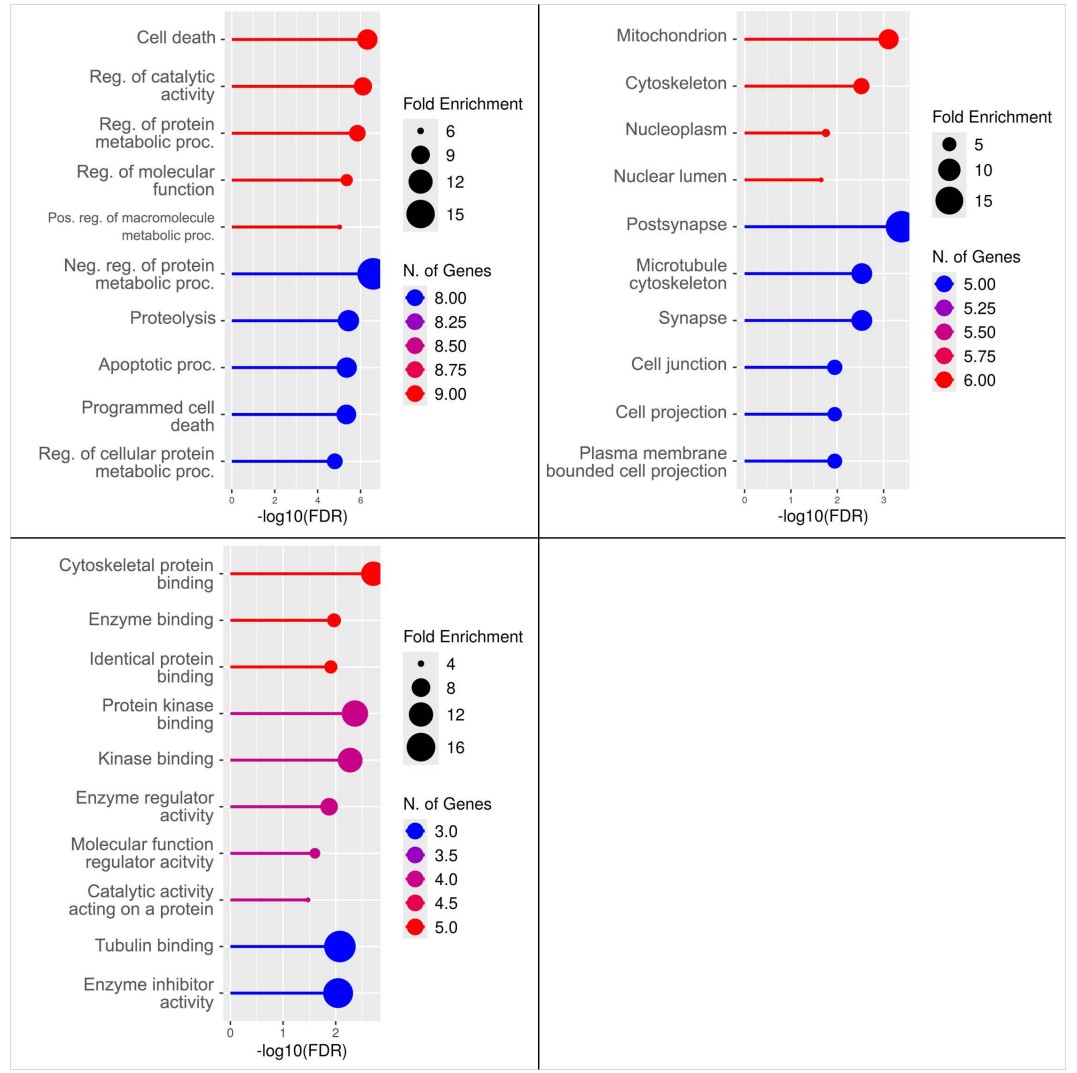

**Fig 6. Functional enrichment analysis using gene ontology biological process (left), cellular component (center) and molecular function (right) terms.**

previous studies [48,50]. These findings underscore its potential as a central node in mediating the overlap between AD and IR-related pathways.

Further, the miRNAs identified in our study targeted several key genes that play central roles in the overlapping pathophysiology of IR and AD. These genes are embedded in critical molecular networks that mediate inflammation, oxidative stress, apoptosis, and glucose metabolism, hallmarks shared by both conditions. Among them, TNF and NFKB1 are the master regulators of neuroinflammation and peripheral insulin desensitization. Acting through major pathways such as NFκB, p38 MAPK, Akt/mTOR, caspase, nitric oxide, and COX, these genes drive the activation of microglia and astrocytes, promoting the release of pro-inflammatory cytokines like TNF-α and IL-1β, which further exacerbate neurodegeneration in AD [51–53].

Functional annotation of these miRNA-targeted hub genes revealed strong enrichment in GO categories and molecular pathways relevant to AD and IR. GO terms such as cytoskeletal protein binding and regulation of protein metabolic

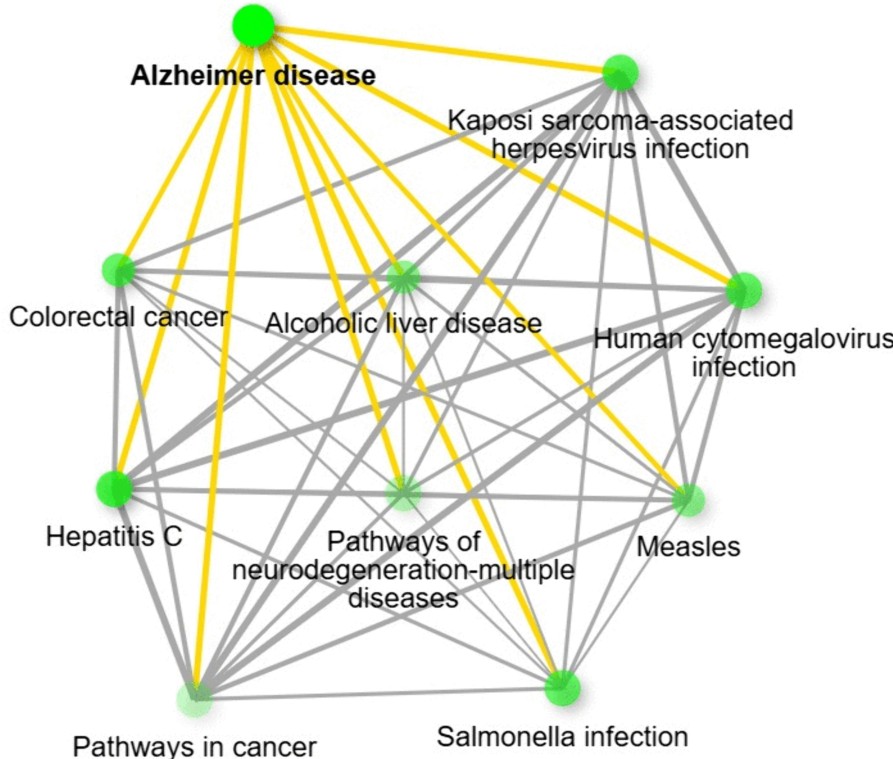

**Fig 7. KEGG network diagram showing terms enriched by top 10 AD-IR hub genes.**

processes implicate genes like CTNNB1 and GSK3B in maintaining synaptic structure, neuronal survival, and protein homeostasis, which are essential for cognitive integrity. CTNNB1 (β-catenin), a key effector of Wnt signaling, intersects with insulin pathways to regulate synaptic plasticity and neuroprotection [54–56]. Enrichment in the term mitochondrial integrity highlights the involvement of genes which are crucial for energy homeostasis and the redox balance. Of particular interest is sα-synuclein (SNCA), a gene traditionally linked to Parkinson's disease but increasingly recognized for its role in AD. Recent evidence indicates that SNCA maintains mitochondrial calcium homeostasis via the mitochondrial-associated ER membrane. Loss of SNCA function impairs this regulatory axis, causing calcium dysregulation, mitochondrial dysfunction, ER stress, and hippocampal neuronal apoptosis, features that align closely with early AD pathology [57].

GSK3B was reaffirmed as a critical molecular bridge between IR and AD, given its involvement in both impaired insulin signaling and tau hyperphosphorylation [54]. Similarly, the apoptotic regulators CASP3 (caspase-3) and CYCS (cytochrome C), both targeted by our miRNA candidates, are central to neuronal loss in AD and β-cell dysfunction in diabetes, further underscoring the shared mechanisms of programmed cell death in these disorders [58,59]. Moreover, enrichment of pathways associated with mitochondrial dysfunction, viral infection, and neuroinflammation suggests that chronic systemic metabolic stress may predispose neurons to degeneration via miRNA-mediated dysregulation of immune and mitochondrial signaling. Our KEGG pathway analysis corroborated these findings, revealing enrichment in the neurodegenerative pathways. Taken together, these findings support a mechanistic framework in which miRNA dysregulation in IR modulates key regulatory genes and convergent molecular pathways, exacerbating neuronal vulnerability and accelerating AD progression.

While bioinformatic analyses provide a robust foundation for identifying candidate miRNAs, experimental validation remains crucial for confirming their biological relevance. Future studies employing both in vitro and in vivo models are

necessary to substantiate the predicted miRNA–target interactions and their functional significance in the context of AD and IR. Validation of miRNA expression patterns in patients with AD exhibiting IR, alongside the assessment of corresponding target gene expression, will offer critical mechanistic insights. Experimental approaches, such as quantitative PCR, luciferase reporter assays, and CRISPR/Cas9-mediated gene editing, are well-suited for elucidating the regulatory roles of the shortlisted miRNAs and their downstream targets. Collectively, this work lays a strong foundation for miRNA-based biomarker development and therapeutic exploration at the intersection of metabolic and neurodegenerative disorders.

## Supporting information

**S1 File. Rationale for inclusion and exclusion of insulin resistance genes.**
(DOCX)

**S2 File. Detailed scoring of miRNAs according to 20 insulin resistance gene targets.**
(XLSX)

**S3 File. Differentially regulated miRNAs across GEO datasets.**
(XLSX)

**S4 File. List of all AD related genes targeted by top-scoring miRNAs.**
(XLSX)

## Author contributions

**Conceptualization:** Tehniat Faraz Ahmed, Farina Hanif.

**Data curation:** Tehniat Faraz Ahmed, Affan Ahmed.

**Formal analysis:** Tehniat Faraz Ahmed, Affan Ahmed.

**Investigation:** Tehniat Faraz Ahmed, Affan Ahmed, Muhammad Bilal Azmi.

**Methodology:** Muhammad Bilal Azmi, Farina Hanif.

**Project administration:** Zeba Haque, Farina Hanif.

**Supervision:** Zeba Haque, Jawwad Us Salam, Farina Hanif.

**Visualization:** Affan Ahmed.

**Writing – original draft:** Tehniat Faraz Ahmed, Affan Ahmed.

**Writing – review & editing:** Muhammad Bilal Azmi, Jawwad Us Salam, Farina Hanif.

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
