## [Decision Letter · Decision Letter 0]

12 Oct 2025

Dear Dr. Hanif,

Thank you for submitting your manuscript to PLOS ONE. After careful consideration, we feel that it has merit but does not fully meet PLOS ONE’s publication criteria as it currently stands. Therefore, we invite you to submit a revised version of the manuscript that addresses the points raised during the review process.

We look forward to receiving your revised manuscript.

Kind regards,

Simone Agostini, Ph.D.

Academic Editor

PLOS ONE

Journal Requirements:

Reviewers' comments:

Reviewer's Responses to Questions

**Comments to the Author**

1. Is the manuscript technically sound, and do the data support the conclusions?

Reviewer #1: Yes

Reviewer #2: Yes

2. Has the statistical analysis been performed appropriately and rigorously?

Reviewer #1: Yes

Reviewer #2: Yes

3. Have the authors made all data underlying the findings in their manuscript fully available?

Reviewer #1: Yes

Reviewer #2: Yes

4. Is the manuscript presented in an intelligible fashion and written in standard English?

Reviewer #1: Yes

Reviewer #2: Yes

Reviewer #1: The manuscript by Ahmed et al. presents a timely and computationally rigorous investigation into the potential role of miRNAs as molecular links between Insulin Resistance (IR) and Alzheimer's Disease (AD). The study's strength lies in its systematic, multi-step bioinformatics pipeline that integrates predictions from multiple algorithms with experimental validation databases. The identification of high-confidence miRNAs like miR-15a-5p, miR-424-5p, and the subsequent network and functional enrichment analysis of AD hub genes provides a valuable resource for the field. The topic is of significant interest given the established epidemiological links between metabolic disorders and neurodegeneration. However, while the computational approach is sound, the study currently remains a purely in silico prediction. Several major concerns regarding biological interpretation, methodological transparency, and contextualization of findings must be addressed before the manuscript can be considered for publication.

1. The manuscript would benefit from a clearer statement of the objectives and hypotheses. Explicitly articulate what knowledge gap the study addresses.

2. The most significant limitation is the absence of any experimental validation for the core predictions. The authors have expertly leveraged existing databases, but the claim that these miRNAs "offer potential targets for early diagnosis and intervention" is premature without confirming their differential expression in relevant biological samples (e.g., serum, CSF, or brain tissue from AD patients with/without comorbid IR/T2DM). The discussion of future experiments (lines 345-354) is appreciated but does not substitute for preliminary validation. I strongly recommend the authors attempt to access publicly available miRNA expression datasets (e.g., from GEO) related to AD and IR to provide at least a preliminary correlation between their predictions and real-world expression data.

3. The selection of the 20 IR-related genes is a critical foundation of the entire study. While the authors mention a "scoping approach" and reference two reviews, a more detailed justification is required.

• A table in the supplementary materials listing these 20 genes with a brief rationale for their inclusion (e.g., "core insulin signaling component," "key regulator of hepatic gluconeogenesis") would significantly strengthen the manuscript.

• The authors should comment on potential selection bias. Were other well-established IR genes (e.g., PPARG, SLC2A2, IL6) considered and excluded? If so, why?

4. The methodology and results for selecting the final 16 miRNAs are somewhat convoluted.

• It is unclear how the T-score, which is an average across all 20 genes, leads to a specific list of miRNAs. Table 3 shows miRNAs with scores for each gene. Does this mean a miRNA is included in the final list if its average T-score across all genes is >1.0? This needs to be explicitly stated in the methodology.

• Furthermore, the biological significance of a miRNA having a high score because it targets many genes weakly versus targeting a few key genes strongly is not discussed. A supplementary figure showing the network of the top 16 miRNAs and their connections to the 20 IR genes would be highly informative.

5. Some important publications in this area of study are missed in this manuscript. For example:

https://doi.org/10.1016/j.ijbiomac.2025.146250

https://doi.org/10.1016/j.prp.2023.155007

Reviewer #2: This article presents a well-designed and scientifically significant computational study aimed at identifying microRNAs (miRNAs) that mediate the crosstalk between insulin resistance (IR) and Alzheimer's disease (AD).The authors utilize a multi-step bioinformatic pipeline, starting with the selection of key IR-related genes, followed by miRNA prediction using five computational tools and validation against two experimental databases. Through a novel scoring system, they shortlist 16 high-confidence miRNAs and subsequently identify their target hub genes involved in AD pathways using network and functional enrichment analyses.However, to further enhance the scientific rigor and impact of the manuscript, several issues outlined below should be addressed.

1. P6 Line102, “A literature search was conducted to ... at both proximal and distal levels.”

Issue:The foundation of this entire study rests upon the initial selection of "20 key IR-related genes." However, the manuscript's methodology for choosing these specific genes is not sufficiently transparent. The text mentions using a "scoping approach" and prioritizing "established modulators," but these descriptions are too general. This lack of specific, objective criteria makes the selection process difficult to evaluate and hinders the study's reproducibility.

Recommendation:It is strongly recommended that the authors elaborate on the concrete criteria used for this gene selection in the Methods section. For instance, were the genes chosen based on: Their inclusion in specific, highly-cited review articles on insulin resistance? (If so, please provide the citations)and so on.

2. P7 Line139, “Based on this scoring framework, … and experimental validation”.

Issue: The authors have developed a novel T-score system to prioritize miRNAs and have set a threshold of "T-score > 1.0" to select candidates for downstream analysis. While the scoring system itself is well-described, the rationale for choosing the specific cutoff value of 1.0 is absent. As this threshold is a critical parameter that directly determines which miRNAs are included in the final analysis, its selection appears arbitrary without a clear justification.

Recommendation: Please provide a rationale for selecting the T-score > 1.0 threshold in the Methods section. If this cutoff was chosen empirically, briefly explain the reasoning behind it. For example, you could state that this value was found to provide an optimal balance between capturing a manageable number of high-confidence candidates and ensuring high predictive and experimental support. Justifying this key parameter is essential for strengthening the study's methodological rigor.

3. P16 Line222,”We then constructed a protein-protein ...are displayed in Figure 3”.

Issue:The central finding of the article is the identification of a set of miRNAs that potentially link IR and AD by targeting key hub genes. However, this core conclusion is not directly and intuitively visualized in any of the figures. While Figure 3 effectively displays the protein-protein interaction network among the hub genes, it critically omits the regulatory miRNAs. As a result, the reader cannot immediately grasp the final predicted miRNA-hub gene regulatory network, which represents the ultimate output of the authors' comprehensive analysis.

Recommendation: It is strongly recommended that the authors create and include a new network visualization. This figure should explicitly illustrate the regulatory relationships between the 16 shortlisted high-scoring miRNAs and the 10 AD hub genes they are predicted to target. Such a figure would serve as a powerful and direct visual summary of the paper's primary contribution and would significantly enhance the clarity and impact of the findings.

**Do you want your identity to be public for this peer review?** For information about this choice, including consent withdrawal, please see our Privacy Policy

Reviewer #1: No

Reviewer #2: **Yes: ** Weilong Yang

---

## [Author Response · Author response to Decision Letter 1]

21 Nov 2025

Reviewer 1:

Comment 1: The manuscript by Ahmed et al. presents a timely and computationally rigorous investigation into the potential role of miRNAs as molecular links between Insulin Resistance (IR) and Alzheimer's Disease (AD). The study's strength lies in its systematic, multi-step bioinformatics pipeline that integrates predictions from multiple algorithms with experimental validation databases. The identification of high-confidence miRNAs like miR-15a-5p, miR-424-5p, and the subsequent network and functional enrichment analysis of AD hub genes provides a valuable resource for the field. The topic is of significant interest given the established epidemiological links between metabolic disorders and neurodegeneration. However, while the computational approach is sound, the study currently remains a purely in silico prediction. Several major concerns regarding biological interpretation, methodological transparency, and contextualization of findings must be addressed before the manuscript can be considered for publication.

Response 1: We thank the reviewer for their generous comments and will make effort to address all concerns to enhance the quality of our research.

Comment 2: The manuscript would benefit from a clearer statement of the objectives and hypotheses. Explicitly articulate what knowledge gap the study addresses.

Response 2: Thank you for the insightful comment. We have revised the introduction (lines 85-95) to include clear statements on objectives and hypothesis and have explicitly articulated the knowledge gap in lines 85-90.

Comment 3: The most significant limitation is the absence of any experimental validation for the core predictions. The authors have expertly leveraged existing databases, but the claim that these miRNAs "offer potential targets for early diagnosis and intervention" is premature without confirming their differential expression in relevant biological samples (e.g., serum, CSF, or brain tissue from AD patients with/without comorbid IR/T2DM). The discussion of future experiments (lines 345-354) is appreciated but does not substitute for preliminary validation. I strongly recommend the authors attempt to access publicly available miRNA expression datasets (e.g., from GEO) related to AD and IR to provide at least a preliminary correlation between their predictions and real-world expression data.

Response 3: We thank the reviewer for this insightful comment and fully agree that experimental validation is essential to strengthen the biological relevance of our predictions. In response, we have now incorporated an additional analysis using miRNA expression datasets from GEO, as recommended (lines 179-192). The Results section now reports the differentially expressed miRNAs and details the overlap with our 16 high-confidence candidates (lines 267-274). We hope these additions provide preliminary validation and directly address the reviewer’s concern by demonstrating concordance between our computational predictions and patient-derived expression profiles.

Comment 4: The selection of the 20 IR-related genes is a critical foundation of the entire study. While the authors mention a "scoping approach" and reference two reviews, a more detailed justification is required.

• A table in the supplementary materials listing these 20 genes with a brief rationale for their inclusion (e.g., "core insulin signaling component," "key regulator of hepatic gluconeogenesis") would significantly strengthen the manuscript.

• The authors should comment on potential selection bias. Were other well-established IR genes (e.g., PPARG, SLC2A2, IL6) considered and excluded? If so, why?

Response 4: Thank you for this comment. We realize it is very important to address this issue. We have addressed this issue comprehensively in the revised manuscript as follows:

• We have added a dedicated subsection titled “Selection of Insulin Resistance Pathway Genes for Analysis” in the Methods section. This subsection now provides a detailed explanation of the criteria and rationale used for selecting the 20 IR-related genes (lines 106-132).

• We have explicitly acknowledged and discussed the potential for selection bias and justified our focus on the canonical PI3K–AKT–mTOR metabolic arm of insulin signaling. The exclusion of certain well-established modulators such as PPARG, IL6, and SLC2A2 was intentional, as these genes influence insulin sensitivity indirectly—through transcriptional, inflammatory, or tissue-specific mechanisms—rather than through direct propagation of insulin receptor signaling within the canonical PI3K–AKT–mTOR axis.

• We have added a new supplementary table (S1) which provides a concise summary of the genes analyzed, their biological functions, and the rationale for inclusion or exclusion.

Comment 5: The methodology and results for selecting the final 16 miRNAs are somewhat convoluted. It is unclear how the T-score, which is an average across all 20 genes, leads to a specific list of miRNAs. Table 3 shows miRNAs with scores for each gene. Does this mean a miRNA is included in the final list if its average T-score across all genes is >1.0? This needs to be explicitly stated in the methodology.

Response 5: We thank the reviewer for highlighting this point and agree that clarification was needed. We have now explicitly stated in the Methods section that a miRNA was included in the final list if its average T-score across all 20 insulin-resistance–related genes exceeded 1.0 (lines 167-178). The revised text also clarifies that the T-score reflects both the breadth of predicted regulation (number of IR genes targeted) and the strength of supporting evidence (cross-tool and experimental validation). The cutoff value was determined empirically based on exploratory analysis of score distributions to retain high-confidence, broadly supported candidates.

Comment 6: Furthermore, the biological significance of a miRNA having a high score because it targets many genes weakly versus targeting a few key genes strongly is not discussed. A supplementary figure showing the network of the top 16 miRNAs and their connections to the 20 IR genes would be highly informative.

Response 6: A new Figure 3 has been included to depict the interaction strengths between IR genes and the shortlisted microRNAs (line 263).

Comment 7: Some important publications in this area of study are missed in this manuscript. For example:

• https://doi.org/10.1016/j.ijbiomac.2025.146250

• https://doi.org/10.1016/j.prp.2023.155007

Response 7: The recommended publications have been added to the manuscript.

Reviewer 2:

Comment 1: This article presents a well-designed and scientifically significant computational study aimed at identifying microRNAs (miRNAs) that mediate the crosstalk between insulin resistance (IR) and Alzheimer's disease (AD).The authors utilize a multi-step bioinformatic pipeline, starting with the selection of key IR-related genes, followed by miRNA prediction using five computational tools and validation against two experimental databases. Through a novel scoring system, they shortlist 16 high-confidence miRNAs and subsequently identify their target hub genes involved in AD pathways using network and functional enrichment analyses. However, to further enhance the scientific rigor and impact of the manuscript, several issues outlined below should be addressed.

Response 1: We thank the reviewer for their generous comments and will make effort to address all concerns to enhance the quality of our research.

Comment 2: P6 Line102, “A literature search was conducted to ... at both proximal and distal levels.”

Issue: The foundation of this entire study rests upon the initial selection of "20 key IR-related genes." However, the manuscript's methodology for choosing these specific genes is not sufficiently transparent. The text mentions using a "scoping approach" and prioritizing "established modulators," but these descriptions are too general. This lack of specific, objective criteria makes the selection process difficult to evaluate and hinders the study's reproducibility.

Recommendation: It is strongly recommended that the authors elaborate on the concrete criteria used for this gene selection in the Methods section. For instance, were the genes chosen based on: Their inclusion in specific, highly-cited review articles on insulin resistance? (If so, please provide the citations)and so on.

Response 2: We thank the reviewer for highlighting this important point. In response, we have added a dedicated subsection titled “Selection of Insulin Resistance Pathway Genes for Analysis” in the Methods section (lines 106-132). This subsection now clearly outlines the objective criteria used for gene selection, based on a structured, literature-driven approach drawing from peer-reviewed studies and authoritative review articles on insulin signaling and insulin resistance (references 5, 27–33). The revised text defines specific inclusion parameters—such as recurrent reporting across multiple sources and representation of functionally distinct components within the canonical PI3K–AKT–mTOR signaling pathway—and provides complete details in the newly added Supplementary Table S3 to enhance transparency and reproducibility.

Comment 3: P7 Line139, “Based on this scoring framework, … and experimental validation”.

Issue: The authors have developed a novel T-score system to prioritize miRNAs and have set a threshold of "T-score > 1.0" to select candidates for downstream analysis. While the scoring system itself is well-described, the rationale for choosing the specific cutoff value of 1.0 is absent. As this threshold is a critical parameter that directly determines which miRNAs are included in the final analysis, its selection appears arbitrary without a clear justification.

Recommendation: Please provide a rationale for selecting the T-score > 1.0 threshold in the Methods section. If this cutoff was chosen empirically, briefly explain the reasoning behind it. For example, you could state that this value was found to provide an optimal balance between capturing a manageable number of high-confidence candidates and ensuring high predictive and experimental support. Justifying this key parameter is essential for strengthening the study's methodological rigor.

Response 3: Thank you for this valuable observation. We have now clearly mentioned a rationale for selection of the T-score > 1.0 cutoff in the Methods section (Lines 167-178). The revised text now explains that this threshold was determined empirically after exploratory analysis of score distributions, which showed that miRNAs with T-scores above 1.0 were consistently supported by multiple prediction tools and experimental evidence while also targeting several genes within the insulin-signaling network. This cutoff thus prevented inclusion of weakly supported interactions.

Comment 4: P16 Line222,”We then constructed a protein-protein ...are displayed in Figure 3”.

Issue: The central finding of the article is the identification of a set of miRNAs that potentially link IR and AD by targeting key hub genes. However, this core conclusion is not directly and intuitively visualized in any of the figures. While Figure 3 effectively displays the protein-protein interaction network among the hub genes, it critically omits the regulatory miRNAs. As a result, the reader cannot immediately grasp the final predicted miRNA-hub gene regulatory network, which represents the ultimate output of the authors' comprehensive analysis.

Recommendation: It is strongly recommended that the authors create and include a new network visualization. This figure should explicitly illustrate the regulatory relationships between the 16 shortlisted high-scoring miRNAs and the 10 AD hub genes they are predicted to target. Such a figure would serve as a powerful and direct visual summary of the paper's primary contribution and would significantly enhance the clarity and impact of the findings.

Response 4: We thank the reviewer for this valuable suggestion. In response, we have added Figure 4b, which illustrates the protein–protein interaction network of Alzheimer’s disease (AD) hub genes identified through STRING, together with their predicted interactions with the shortlisted IR-associated microRNAs (lines 293-295). This figure provides an integrated visualization of the molecular crosstalk between key AD hub genes and their regulatory miRNAs.

---

## [Decision Letter · Decision Letter 1]

27 Nov 2025

Mapping the microRNA-mediated crosstalk between insulin resistance and Alzheimer’s disease: a computational genomic insight

PONE-D-25-37575R1

Dear Dr. Hanif,

We’re pleased to inform you that your manuscript has been judged scientifically suitable for publication and will be formally accepted for publication once it meets all outstanding technical requirements.

Kind regards,

Simone Agostini, Ph.D.

Academic Editor

PLOS ONE

Additional Editor Comments (optional):

Reviewers' comments:

Reviewer's Responses to Questions

**Comments to the Author**

Reviewer #1: All comments have been addressed

Reviewer #2: All comments have been addressed

2. Is the manuscript technically sound, and do the data support the conclusions?

Reviewer #1: Yes

Reviewer #2: Yes

3. Has the statistical analysis been performed appropriately and rigorously?

Reviewer #1: Yes

Reviewer #2: Yes

4. Have the authors made all data underlying the findings in their manuscript fully available?

Reviewer #1: Yes

Reviewer #2: Yes

5. Is the manuscript presented in an intelligible fashion and written in standard English?

Reviewer #1: Yes

Reviewer #2: Yes

Reviewer #1: The authors have comprehensively addressed all of the concerns I raised in my initial review. They have provided detailed, point-by-point responses to each comment and have implemented extensive revisions to the manuscript that reflect this.

Reviewer #2: (No Response)

**Do you want your identity to be public for this peer review?** For information about this choice, including consent withdrawal, please see our Privacy Policy

Reviewer #1: No

Reviewer #2: **Yes: ** Weilong Yang

---

## [Editor Report · Acceptance letter]

PONE-D-25-37575R1

PLOS ONE

Dear Dr. Hanif,

I'm pleased to inform you that your manuscript has been deemed suitable for publication in PLOS ONE. Congratulations! Your manuscript is now being handed over to our production team.

Kind regards,

on behalf of

Dr. Simone Agostini

Academic Editor

PLOS ONE